# Relationship between Pulp–Tooth Area Ratio and Chronological Age among Saudi Arabian Adults: A Cone Beam Computed Tomography Image Analysis

**Abdullah Alqarni [1], Muhammed Ajmal [1,*], Reem Mohammed Hakami [2], Abeer Abdullah Alassmi [2], Sandeepa Nuchilakath Chalikkandy [1] and Saeed Arem [1]**

1   Department of Diagnostic Sciences-Oral Biology, College of Dentistry, King Khalid University, Abha 62529, Saudi Arabia
2   College of Dentistry, King Khalid University, Abha 62529, Saudi Arabia
*   Correspondence: drmajmal@kku.edu.sa

**Abstract:** The pulp–tooth area ratio (PTR) from radiographic dental archives is considered useful for age estimation in the forensic field. Since there have been no studies conducted in the Saudi Arabian population using the PTR method, this study aimed to assess the relationship between the pulp–tooth area ratio of maxillary canines and central incisors and chronological age among Saudi Arabian adults using CBCT and to compare the selected teeth's predictive power. For this, 100 CBCT scans of 62 male and 38 female patients aged between 20 and 60 years were assessed using OnDemand 3D DentalTM software (Cybermed Co., Tustin, CA, USA) in the axial and sagittal section of each tooth. The mean age estimated using the PTR of the maxillary incisors (39.98 years) was higher than the canines' (37.9 years). A negative correlation coefficient 'r' value was seen between PTR and age. The strongest inverse correlation was noted between age and incisor axial view (0.47) ($p < 0.05$), Maxillary central incisors had higher $R^2$ in both genders (males: PTRS-0.18, PTRA-0.17; females: PTRS-0.19, PTRA-0.35) than canines. We conclude a weak correlation between the pulp–tooth ratio and chronological age estimation. CBCT pulp–tooth ratios of axial and sagittal sections of incisors were inconclusive in estimating the age of Saudi Arabian individuals owing to their low coefficient of estimation. Among the tooth types studied, maxillary incisors were better predictors than canines.

**Keywords:** age estimation; cone-beam computed tomography; forensic odontology; secondary dentin deposition

## 1. Introduction

Age estimation is a challenging task under many circumstances in the global socio-political scenario [1–3]. Teeth are more resistant to physiochemical changes caused by intrinsic and extrinsic factors in living and deceased persons. This makes teeth a better choice than other skeletal features for age prediction [4–8]. The reliability of age estimation using teeth has been of great interest among researchers recently. Although several dental-based age estimation methods are available, many of them are time-consuming, require extraction of particular teeth, and expensive laboratory equipment [9,10]. Radiological advances in dentistry have paved the way to investigate age without the need for tissue harvesting [11–13]. Although radiological exposure in living beings raises ethical concerns, radiographic dental archives are an excellent source for predicting age estimation [8,14].

For the last several years, age estimation by the pulp–tooth ratio (PTR) has been considered a valid, noninvasive, and cost-effective technique [15–19]. Interestingly, numerous radiological investigations using two-dimensional imaging techniques revealed a correlation between age and pulp–tooth ratio [18]. Most of these studies that utilized two-dimensional imaging focus only on the field of vision while relatively blurring up

the remaining portions. Thus, it is essential to re-evaluate and validate the age estimation using the pulp–tooth ratio with three-dimensional radiological measures [20].

Multiple pieces of evidence using three-dimensional imaging to estimate age demonstrated that PTR is a valuable tool for age prediction [21–24]. For example: a study was carried out in an Egyptian population by Afify et al. [21] using the principles of Cameriere et al. [25] to assess the pulp and tooth areas. This study assessed 500 panoramic radiographs of individuals from Egypt. These were from 262 males and 238 females with the age ranging from 18 to 71 years. Using AutoCAD 2010, the PTR was estimated. The collected data underwent correlation and regression analysis, showing a high correlation with age (r = −0.95 in terms of second premolar and canine (r = −0.94)). On the other hand, the first premolar displayed a lesser degree of correlation (r = −0.91). Based on the findings of this study, it was concluded that this method is a valuable approach for reasonably accurate age assessment.

In a separate investigation conducted by Sasaki et al. [22] among the Japanese population, it was observed that the pulp volume decreases over time. A Bayesian approach was proposed for estimating human age in the canines of the mandibular arch. A likelihood function was generated for pulp volume ratio by fitting the model and a probability density function was estimated for a given age based on a given PVRrt.

Vandevoort et al. [24] conducted another study in a Belgian population to establish a connection between dental age and an individual's chronological age. An X-ray microfocus computed tomography unit (microCT) with a spatial resolution of 25 µm was employed. This scanning technique was applied to examine 43 single-root extracted teeth obtained from 25 individuals with well-documented chronological ages. Two examiners employed custom-made analysis software to derive numerical measurements for both pulpal and tooth volumes. Subsequently, the ratio between the two volumes was calculated and subjected to statistical analysis. The study revealed no significant differences either within or between examiners, indicating a high level of agreement. When performing linear regression analysis, a low coefficient of determination (0.31) was obtained. Despite being somewhat time-consuming, this method demonstrates promising outcomes for assessment of dental age when utilizing X-ray microfocus computed tomography.

Thus, advancements in three-dimensional imaging opened up opportunities to incorporate volumetric measurements and provide a more comprehensive understanding of the ongoing process of tooth mineralization [20]. The introduction of cone beam computed tomography (CBCT), specifically designed for the maxillofacial region, has expanded the capabilities of three-dimensional imaging. This novel technique now enables the acquisition of volumetric data in vivo. These acquisitions present promising prospects for exploring and analyzing volumetric measurements of teeth. Subsequently, many researchers shifted the focus towards the use of cone beam computed tomography (CBCT) and found it to be ideal and accurate for the PTR measurement method [19,20,26]. For example, Salami et al. [19] estimated the chronological age of Iranian individuals using CBCT based on Kvaal's method [17] to analyze the correlation between age and morphological variables of the maxillary canine teeth. The morphological variables were measured in different planes. Significant negative correlations were observed between age and all the measured variables. There were no notable differences in the mean variables between males and females. The regression model exhibited a correlation of 0.88 between the estimated age and the actual age. Based on the findings, the developed equation provides a reasonably accurate estimation of individuals' age, with an allowable error of less than five years.

In a Belgian population, [23], Star et al. assessed a method for estimating human dental age by examining the ratio between the pulp volume and the corresponding tooth volume, using CBCT images of anterior teeth. A total of 111 CBCT images obtained from clinical scans (Scanora® 3D dental cone beam unit) were utilized in this study. Using Simplant® Pro software, the pulp–tooth volume ratio was calculated for 64 incisors, 32 canines, and 15 premolars based on the three-dimensional images. A linear regression

model was applied here and among the teeth examined, the incisors displayed the best correlation between the pulp–tooth volume ratio and age.

Age determination is a key factor in solving crimes, identification of dead bodies, issues related to physical assault, illegal immigration, etc., in every country. However, there are no available studies in the Saudi Arabian population using the above-mentioned advantages of CBCT to correlate the PTR of the anterior teeth (central incisor and canine) with the chronological age. This is the reason why we chose to conduct this research. Among various teeth in the dental arch, maxillary incisors and canine teeth have a more prominent volume of pulp, less wear from the diet, high survival rates, and ease of performing radiographic measurements, and the reliability of these teeth are reported in previous studies [3,8,27–31]. Therefore, this study aimed to assess the relationship between the pulp–tooth area ratio of the maxillary canines and central incisors and chronological age among Saudi Arabian adults using CBCT.

## 2. Materials and Methods

### 2.1. Ethical Considerations

The protocol for the study and collection of radiographs of human subjects was approved by the Institutional review board at King Khalid University (KKU) (IRB/KKUCOD/ ETH/2021-22/049).

Assuming a power of 90% with a confidence interval of 95% to obtain correlation confidence of 0.3, a sample size of 100 was required. Hence, in our study, 100 samples, i.e., 100 central incisors and 100 canine teeth, were chosen using a simple random technique from the CBCT archives of the Department of Oral and Maxillofacial Radiology, College of Dentistry, KKU, Saudi Arabia between January 2019 and December 2021.

### 2.2. Inclusion and Exclusion Criteria

In our study, CBCT scans of patients of both sexes ranging in age from 20 to 60 years that had information about the patient's date of birth, gender, and the presence of at least one maxillary central incisor and canine with fully formed roots were included. Teeth with any dental disease/conditions, malformations, and restorations were excluded. Further, radiological images with poor-quality artifacts were not analyzed.

### 2.3. Methods

2.3.1. CBCT Acquisition

The CBCT images in the study were obtained using Kavo 3D Pro with a standard setting of 89 kV and 8 mA. Further, the field of view was selected as $8 \times 8$ cm with a resolution of 0.2 mm. The software program OnDemand 3D Dental software (Cybermed Co., Tustin, CA, USA) assessed the pulp–tooth ratio on a computer screen under standardized viewing conditions of each tooth in the axial and sagittal section.

2.3.2. Measurement

The tooth and pulp areas were measured in triplicates, and their averages were used in the final computations. The examiners opened the qualified CBCT samples on the desktop using the software. If needed, initial brightness, contrast, and sharpness adjustments were made to obtain better results. In the sagittal plane, the pulp/teeth ratio (PTR) longitudinal axis of the tooth was measured from the crown tip to the root apex. Next, using a cross-sectional plan, a midsagittal section with the largest tooth and pulp areas was selected. Finally, the toolbar was used to trace the total tooth area using a set of points, as shown in Figure 1. The contour of the pulp from the crown to the root apex was mapped, and the area of the pulp was measured subsequently. Finally, PTR in the sagittal plane was measured by dividing the pulp's area into the tooth's area (Figure 1).



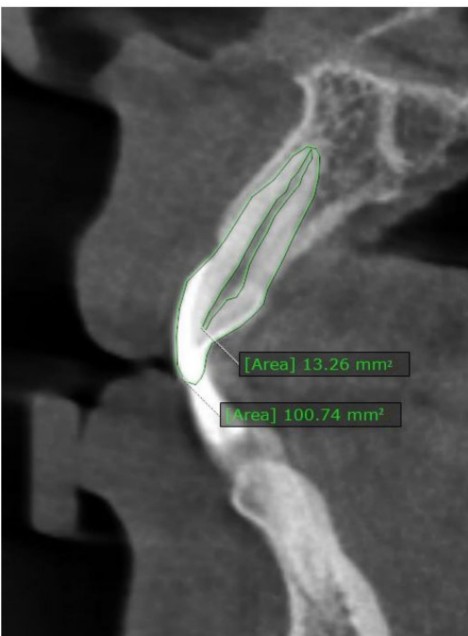

**Figure 1.** PTR ratio in sagittal plane is 0.13.

The tooth's longitudinal axis from the crown tip to the root apex was measured to determine PTR in the axial plane. The axial plane of the software utilized the cementoenamel junction of the respective tooth as the reference point, as shown in Figure 2. The authors used points to measure the tooth and pulp areas, and the PTR axial was computed. A recent systematic review endorsed that there are no significant differences between permanent teeth from the left and right sides of the jaw. Hence, we chose teeth from the left or the right side best suited for measurement [18].

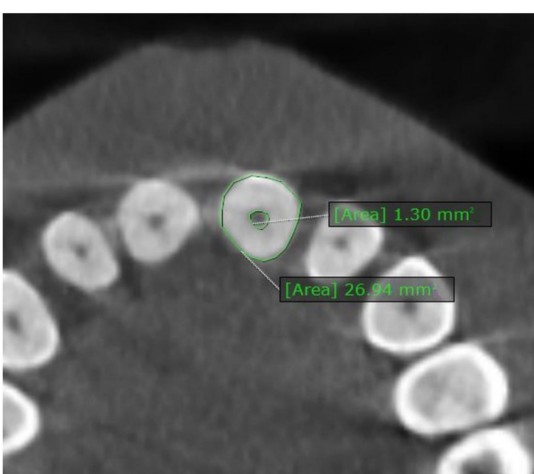

**Figure 2.** PTR ratio in axial plane is 0.05.

### 2.3.3. Examiner Calibration

The evaluations were carried out by two trained dental practitioners (MA and SN) with prior skills in CBCT image interpretation and the use of the software program. In order to overcome observer bias, two examiners assessed the PTR separately. Before the start of the study, two examiners evaluated 20 CBCT images for PTR in two sections, i.e., axial and sagittal. These sections were not included in the study sample. Kappa coefficient for inter-examiner reliability was found to be good (Kappa statistic—0.84).

### 2.4. Statistical Analysis

The data were computed using SPSS, version 17 (SPSS Inc., Chicago, IL, USA), and a *p* value less than 0.05 was considered statistically significant. Unpaired *t*-test was used to assess the difference between PTR in both the planes of males and females. Pearson's correlation test was used to assess the correlation between the age and PTR. To predict the age of an individual simple linear regression analysis was used. To assess the Inter observer variability, Kappa statistics was used.

### 3. Results

The total of 100 CBCT images included 62 male and 38 female patients. The mean age and PTR of the sampled teeth in the axial and sagittal section are mentioned in Table 1. Gender-wise PTR was compared in both the planes and shown in Table 2; the *t*-test revealed a non-significant difference in both the incisors and canines. The incisors' overall mean PTR (0.08) was higher than the canines' (0.06) in the axial plane ($p < 0.001$).

**Table 1.** Descriptive statistics of the study parameters.

| Parameter | Incisors | Canines | *p* Value |
|---|---|---|---|
| Age | 39.53 ± 10.99 | 39.53 ± 10.99 | |
| PAS | 14.64 ± 3.40 | 19.55 ± 22.58 | |
| TAS | 115.15 ± 12.90 | 133.65 ± 22.58 | |
| **PTRS** | 0.12 ± 0.02 | 0.14 ± 0.02 | $p < 0.001$ ** |
| PAA | 2.52 ± 0.88 | 2.40 ± 0.84 | |
| TAA | 32.13 ± 4.42 | 38.27 ± 6.36 | |
| **PTRA** | 0.08 ± 0.02 | 0.06 ± 0.02 | $p < 0.001$ ** |

PAS—Pulpal area in sagittal plane, TAS—tooth area in sagittal plane, PTRS—pulp to tooth area in sagittal plane, PAA—pulpal area in axial plane, TAA—tooth area in axial plane, PTRA—pulp to tooth area in axial plane; **—highly significant.

**Table 2.** Gender-wise comparison of PTR in two planes.

| Teeth | Plane | Females (n = 38) | Males (n = 62) | *p* Value |
|---|---|---|---|---|
| **Incisors** | PTRS | 0.12 ± 0.02 | 0.13 ± 0.02 | 0.18, NS |
| | PTRA | 0.07 ± 0.02 | 0.08 ± 0.02 | 0.20, NS |
| **Canines** | PTRS | 0.15 ± 0.03 | 0.14 ± 0.02 | 0.20, NS |
| | PTRA | 0.06 ± 0.01 | 0.06 ± 0.02 | 0.65, NS |

NS—non-significant.

Table 3 depicts the correlation coefficient between age and PTR based on teeth and section viewed. The negative 'r' value was demonstrated since the pulp–tooth area ratio of incisors and canine teeth decreased as the age increased.

**Table 3.** Correlation coefficient between chronological age and PTR.

| Tooth | Plane | Overall | | Male | | Female | |
|---|---|---|---|---|---|---|---|
| | | r | *p* | r | *p* | r | *p* |
| Incisor | PTRS | −0.42 | 0.001 ** | −0.42 | 0.001 ** | −0.44 | 0.005 * |
| | PTRA | −0.47 | 0.001 ** | −0.41 | 0.001 ** | −0.59 | 0.001 ** |
| Canine | PTRS | −0.18 | 0.09, NS | −0.08 | 0.58, NS | −0.32 | 0.07, NS |
| | PTRA | −0.30 | 0.004 * | −0.37 | 0.005 * | −0.22 | 0.22, NS |

r—correlation coefficient, *—significant, **—highly Significant, NS—non-significant.

A statistically significant correlation between PTR and age was demonstrated for all teeth examined except for canine in sagittal view. Among the teeth examined in this study comparatively strongest (low negative) inverse correlation was noted between age and incisor axial view (0.47), while the negligible correlation was reported in the canine's sagittal plane (0.18). Gender-wise correlation demonstrated both males and females had higher negative 'r' values for incisors than canines. Although a statistically significant correlation ($p < 0.05$) was observed for incisor teeth in either gender, the sagittal sections of males had higher r values, i.e., r = males −0.42 versus females −0.44, whereas in the axial view, females reported higher values, i.e., r = males −0.41 versus females −0.59. The canine tooth was reported as a weak predictor for age estimation in the sagittal plane. However, the axial view demonstrated a correlation for the male gender ($p = 0.005$).

The higher the score of $R^2$ and the lesser the SEE, the greater the accuracy of estimating the age. Table 4 demonstrates the regression equation for age estimation in overall sample of both the planes of incisors and canines. Table 5 demonstrates that incisors had higher $R^2$ in both genders (Males: PTRS = 0.18, PTRA = 0.17; Females: PTRS = 0.19, PTRA = 0.35) than canines. Notably, the sagittal section of the incisor's PTR had higher $R^2$ values in males, whereas in females, axial PTR values were higher. The lesser the SEE, the greater the precision toward chronological age. In our study, males reported an error in the range of nine years for both incisors and canines in axial view, whereas in canines a sagittal view error of approximately ten years was noted. Based on Table 5, the SEE in females were higher with almost eleven years for canines compared to approximately ten years in incisors. Table 6 shows the mean estimated age using the regression equation in both the planes in incisors and canines.

**Table 4.** Regression equation for age estimation.

| Tooth | Plane | Equation | $R^2$ | SEE | *p*-Value |
|---|---|---|---|---|---|
| Incisor | PTRS | Age = 64.082 − 189.696 (PTRS) | 0.17 | 10.12 | 0.001 ** |
| Incisor | PTRA | Age = 57.667 − 223.549 (PTRA) | 0.22 | 9.82 | 0.001 ** |
| Canine | PTRS | Age = 49.430 − 79.504 (PTRS) | 0.03 | 10.94 | 0.096, NS |
| Canine | PTRA | Age = 48.534 − 173.857(PTRA) | 0.09 | 10.60 | 0.004 * |

*—significant, **—highly significant, NS—non-significant.

**Table 5.** Gender-wise regression equation for age estimation.

| Gender | Tooth | Plane | Linear Regression Equation | $R^2$ | SEE | *p* Value |
|---|---|---|---|---|---|---|
| Males | Incisor | PTRS | Age = 67.530 − 207.669 (PTRS) | 0.18 | 9.80 | 0.001 ** |
| | | PTRA | Age = 55.986 − 189.092 (PTRA | 0.17 | 9.84 | 0.002 * |
| | Canine | PTRS | Age = 42.508 − 32.975 (PTRS | 0.01 | 10.58 | 0.58, NS |
| | | PTRA | Age = 51.958 − 217.293 (PTRA) | 0.14 | 9.85 | 0.05 * |
| Females | Incisor | PTRS | Age = 61.916 − 186.999 (PTRS) | 0.19 | 10.66 | 0.01 * |
| | | PTRA | Age= 61.821 − 302.434 (PTRA) | 0.35 | 9.58 | 0.001 ** |
| | Canine | PTRS | Age = 60.335 − 159.303 (PTRS) | 0.10 | 11.59 | 0.07, NS |
| | | PTRA | Age= 45.999 − 142.759 (PTRA) | 0.05 | 11.93 | 0.21, NS |

*—significant, **—highly significant, NS—non-significant.

**Table 6.** Estimated age using the regression equation for overall sample.

| Tooth | Plane | Predicted Value | | | SEE |
|---|---|---|---|---|---|
| | | Minimum | Maximum | Mean ± SD | |
| Incisor | PTRS | 29.94 | 52.70 | 39.98 ± 4.60 | 10.12 |
| Incisor | PTRA | 26.37 | 50.96 | 39.98 ± 5.19 | 9.82 |
| Canine | PTRS | 33.52 | 43.86 | 37.94 ± 1.98 | 10.94 |
| Canine | PTRA | 25.93 | 46.79 | 37.94 ± 3.36 | 10.60 |

## 4. Discussion

To our knowledge, this is the first report of chronological age estimation using the pulp–tooth ratio of maxillary central incisors and canine and its predictive power using CBCT in a Saudi Arabian population. Reduction in the pulp chamber is directly proportional to chronological age [32]. Numerous pieces of evidence from different communities gave mixed results for the applicability of the pulp–tooth ratio method of age prediction [8,19,33,34]. Hence, the present study explored the apposition pattern of dentin and the predictive power of maxillary incisors and canines in age estimation among Saudi individuals. Further, the reliability and suitability of the pulp/tooth area ratio tailored regression model for age determination in this population using CBCT teeth images were tested. The results of the study found that the incisors' overall mean PTR was higher than the canines in axial plane only ($p < 0.001$). A negative correlation was demonstrated between the pulp–tooth ratio of incisors and canine teeth and age. Moreover, a significant correlation between PTR and age was established for all teeth examined except for canines in sagittal view.

These findings can be used in forensic science for age estimation. The resistance of the tooth to environmental changes makes it an excellent material for this purpose [5,35]. In adults, the way secondary dentin is laid off was found to be a good age estimation method [33]. However, evidence refutes that dentin deposition, to a larger extent, is influenced by environmental and genetic factors implying the role of racial and cultural factors in age prediction [7,33,36]. Therefore, a customized regression method of age estimation applied to the indigenous population regionally is recommended for estimating chronological age.

Various methods have been experimented with for the reliability of age estimation. Among these methods, radiographic estimation of dental age is gaining keen interest due to its easy applicability, noninvasive technique, and cost-effectiveness [6,18,33]. In this study, measurements of PTR showed no significant gender difference. This result is similar to earlier studies on single-rooted teeth [12,37,38]. A study by Nemsi [12] reported the assessment of chronological age by taking into account the pulp/dentin area ratio at cervical region of the anterior teeth. It was found that dentin deposition on canine and premolar had a very good correlation with age (r = −0.83 and −0.83, respectively). Another study by Dehghani et al. [37] used digital panoramic radiographs and AutoCAD software to estimate age from the pulp measurements of mandibular and maxillary canines. The results showed a significant but inverse correlation between these parameters. The values of upper canine were more accurate than the lower canine in estimation of age. Jeevan et al. [38] calculated these pulpal measurements using a radiovisiography technique and found that these regression models, when used in individuals below 45 years of age, demonstrated superior outcome in age prediction.

However, the findings of the present study contradict the work by Cameriere et al. and Lee et al. [39,40]. In the study by Cameriere et al. [39], the authors aimed to explore the relationship between the pulp/tooth ratio of incisors and age, using a linear regression model. The findings indicated that the regression equation accounted for a range of variance in age estimation, from 51.3% (in the case of lateral incisors with a mean error of 8.44) to 81.6% (upper lateral incisors with a mean age of 5.34 years). These outcomes suggest that while incisors may be less acceptable than other teeth studied, they can still serve as an

alternative for estimating age-at-death in situations where the latter teeth are unavailable. Lee et al. [40] conducted a study in a Korean population to assess the feasibility of using the pulp/tooth area ratio in lower premolar teeth. A panoramic view was used in this study. Gender-based independent t-tests revealed significant differences. The assumptions of the regression models and the accuracy of prediction in PTR and PrRR (PTR ratio in the root part) were found to be satisfactory. It was concluded that the panoramic radiography-based pulp/tooth area ratio is a promising tool in this area. It is noteworthy that female subjects exhibited greater accuracy compared to male subjects.

Lack of difference in pulp–tooth area in the present population may be an added advantage in conditions where gender estimation is otherwise not feasible. A statistically significant correlation between age and PTR was found in the axial and sagittal planes of maxillary incisors and canines. The axial plane of the maxillary incisor was the most closely correlated with age (r = −0.47), followed closely by sagittal view (r = −0.41); however, the sagittal view of canines revealed the negligible correlation (r = −0.17). Also, the results of our study are similar to the reports by Babshet et al. in the Indian population, who reported similar degrees of correlation between PTR and age [15]. This low correlation may be attributed to the slow and irregularly paced secondary dentine formation in specific populations. Hence, the low or negligible correlation of incisors and canines challenges the reliability of these two teeth as an age predictor.

There are reports of higher 'r' values corroborating excellent correlation coefficient between different tooth types and PTR [18]. However, most of the studies utilized two-dimensional radiographs. Compared to a wide range of 'r' values reported by two-dimensional techniques, CBCT imaging would be promisingly precise to validate the tooth as an age estimator [41].

A couple of studies using CBCT imaging in the Turkish and Malaysian population for age prediction reported an 'r' value (correlation coefficient) similar to our study [3,33]. However, a recent meta-analysis by Barbosa et al. for age prediction using pulp–tooth quantification by CBCT technique reported 'r' value of 0.66 for the upper central incisors and 0.59 for upper canines justifying moderate correlation [42]. Our study reported 'r' values, i.e., 0.41 for incisors; 0.17 for canines, which are far less than the value reported in the systematic review. This indicates the inconclusive nature of maxillary central incisors and canines as an age predictor in this Saudi Arabian population. We believe that further studies with combinations of other teeth in this population to find teeth with a strong correlation coefficient, if discovered, would be of utmost use in forensic science.

The coefficient of determination ($R^2$) is a good predictor of the strength of association between two variables. In the present study, among tooth and plane combinations, the axial plane of maxillary incisors had the best $R^2$ value; 0.19 in females and 0.18 in males. As reported by Rai et al. in an Indian population, and in our study, too, only PTR in the axial plane had a significant coefficient of determination for age prediction. Further, our study had lesser error rates than the Indian study [30]. Compared to previous work using computed tomography, we found a compromised co-efficient of determination $R^2$, necessitating us to cautiously utilize teeth as a tool of age determinant in the present population [3,29,43]. Furthermore, as much of the previous evidence either utilized 2D age estimation or CBCT analysis with Cameriere's method, higher $R^2$ values could have been declared. It is noteworthy that although a negative correlation coefficient is reported in the present study, the low $R^2$ value and error rates tell us how far a tooth under study can be relied on as an age predictor. Hence, the literature supporting a specific tooth as an age estimator that lacked reporting $R^2$ and SER (Standard Error Rates) estimation should be cautiously considered.

In our study, incisors precisely estimated the age using regression analysis in both planes, whereas in canines, the axial view was a better predictor than sagittal sections. Haghanifar et al., in their work on the Iranian population using CBCT imaging, also reported maxillary incisors to be a better age predictor along with a weak correlation of canines in the sagittal plane [43]. This clearly demonstrated that tooth type and their

planes were determinants of age estimation. Therefore, the authors recommend studies with population-specific three-dimensional age estimation to analyze the co-efficient of age determination and their error rates.

The present findings of this study open newer vistas for age estimation. Nonetheless, most of the available evidence supports the use of specific teeth as a reliable indicator for age estimation using correlation coefficient values. Estimating $R^2$ (co-efficient of determination) and SE is essential to accurately assess their strength of association in different planes, and using CBCT in specific populations can be useful.

The regression-based method of age prediction in maxillary incisors evaluated an individual's chronological age within the acceptable range set in the field of forensic science, i.e., SE of ten years [21]. However, a more accurate prediction with an error range of less than five years can justify the applicability of this technique.

There were some limitations in our study. Firstly, we did not restrict this technique to teeth in one quadrant of the mouth. Secondly, as different methods (for instance, many studies assessed pulp–tooth volume, and many did not report $R^2$) and radiographic techniques were used in age estimation, we had a limited number of articles to compare and validate. In terms of methodology, convenience sampling was employed, which could be one of the methodological limitations. Finally, data regarding systemic diseases and history of orthodontics treatment could not be elucidated in these patients that could have an influence on the present results.

Robust age estimation techniques are important for any population. This will be useful in forensic settings, to solve medico-legal issues, and in any future disaster management. In order to validate the use of CBCT for dental age estimation, we recommend:

-   Future research studies with larger sample sizes regionally on indigenous populations should be targeted for generalization.
-   Different CBCT scanners and resolutions of the various image sections, planes, and tooth types should be warranted.
-   As forensic experts need precise validation, in addition to correlation coefficient, strength of association and error rates should be computed when assessing particular teeth as an age estimator in specific populations.

## 5. Conclusions

According to the results of our study, a weak correlation existed between chronological age and pulp–tooth ratio. The correlation coefficients were better for maxillary incisors in both genders and planes compared to canines. We found that including gender had no significant effect in age estimation. Age-wise variation in secondary dentine deposition warrants further research in this population to check the applicability of this age-related regression coefficient.

Furthermore, CBCT pulp–tooth ratio along axial and sagittal sections of incisors were inconclusive to estimate the age of Saudi Arabian individuals owing to their low coefficient of estimation. Among the tooth types studied, maxillary incisors were better predictors than canines due to their lesser error range. However, further multicentric studies with different tooth combinations are needed to validate this.

Moreover, the present regression model drafted for the Saudi Arabian population reported a lower co-efficient of determination between pulp–tooth ratio and chronological age estimation. This warrants the use of additional age estimation parameters for reliable prediction.

**Author Contributions:** Conceptualization, M.A., R.M.H. and A.A.A.; methodology, M.A., R.M.H. and A.A.A.; software, M.A., R.M.H., A.A.A. and S.A.; validation, A.A., M.A. and S.A.; formal analysis, M.A. and S.N.C.; investigation, R.M.H. and A.A.A.; resources, A.A.; data curation, M.A., S.N.C. and S.A.; writing—original draft preparation, M.A. and S.N.C.; writing—review and editing, A.A., M.A., S.N.C. and S.A.; visualization, R.M.H. and A.A.; supervision, M.A.; project administration, M.A. All authors have read and agreed to the published version of the manuscript.

**Funding:** This research received no external funding.

**Institutional Review Board Statement:** The research was approved by the research ethics committee, College of Dentistry, King Khalid University, Approval no: IRB/KKUCOD/ETH/2021-22/049.

**Informed Consent Statement:** Patient consent was waived as this is a retrospective radiographic study.

**Data Availability Statement:** Data are available from the corresponding author upon reasonable request.

**Conflicts of Interest:** The authors declare no conflict of interest.

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
