# Peer review of "Relationship between Pulp–Tooth Area Ratio and Chronological Age among Saudi Arabian Adults: A Cone Beam Computed Tomography Image Analysis"

_applsci, doi:10.3390/app13137945_

Round 1
Reviewer 1 Report
The paper is based on outstanding research. However, although the aim of the study is somehow described, what triggered the research is not presented. In this sense, the objective of this study should be clarified in the "Introduction" and "Abstract" Sections. In parallel, the results could be discussed in light of the research objective.
The method is robust, and the illustration of the results is sufficient. The study's added value is high, but it should be highlighted better.
1. It seems that the study attempts to predict human dental age by examining the ratio between the pulp volume and the corresponding tooth volume.CBCT images of anterior teeth answer this purpose.
2. The originality of the study is not highlighted. It seems that the research methodology is drawn from previous similar studies. However, the generic added value is not underlined.
3. The only element that diversifies the study is the employment of maxillary canines and central incisors in the prediction process. This is the study's advantage compared to other studies in the field. Nevertheless, the specific gap that the study addresses needs more clarification.
4. The methodology appears to be robust. The only suggestion is the development of the methodology in a sequence of steps.
5. The conclusion and the discussion of the results is based on the comparison of the study's findings to the ones of similar studies. Though, the results are not fully interpreted according to the study's aim. This is because the study's added value is not highlighted to facilitate such interpretation.
6. The references appear to be relative to the study's objective. The graphical illustration of the results is sufficiently comprehensible. Tables and figures also add to our comprehension.
In my point of view, the main "flaw" of the manuscript is that the study's added value needs to be underlined, and that the results' interpretation should be governed by the generic study's objective.
A slight English proofreading is required.
Reviewer 2 Report
The paper deserve to be published. The authors honestly assum the limit of this research which is not that original generaly speaking but useful as a contribution to the numerous previous research
Reviewer 3 Report
The concept of using pulp tooth ratio (PTR) to estimate the chronological age seems to be a well know concept and there are many studies that have already been published on this topic. There are few studies that use panoramic x-rays and other intra-oral x-rays in Saudi population. Not sure if there is anything new that is being investigated/presented in the study other than using CBCT. However, there are some major concerns about how the PTR was calculated in the study. The estimates in the results don’t really estimate the age correctly. Please refer to the below comments. The paper in the current form needs major revisions.
Abstract:
Line 27-31: The last few lines seem very generic and may need to be rephrased. Data in table #2 show overall incisor and PTRS and PTRA along with canine PTRA are significantly associated with physiological age and it is contradicting the statement “CBCT pulp tooth ratio of axial and sagittal sections of incisors were inconclusive in estimating the age of Saudi Arabian individuals”. Not clear how is coefficient of estimation being measured exactly too. Please describe in methods how is the correlation being classified as low vs high.
Introduction:
Why was pulp volume and tooth volume not considered? Is there an advantage of volume vs area from CBCT?
Methods/Results:
· How was PTR ratio calculated? Isn’t the pulp area always smaller than tooth area? If so, shouldn’t PTR always be less than 1. How come PTR is in the range of 25.93 to 52.74?
· For figure: 1 and 2, please provide the calculation of how PTR was calculated? Based on the numbers in the figure, the ratio is 0.13 and 0.048. But the values in table 1 range from 25.93 to 52.74?
· Line 181: Was the normality of data tested in any way?
· Table 1: The mean values match to the exact decimals for sagittal and axial groups when the SD, min and max differ. Please verify?
· Table 3: What are the p-values for the coefficients/equations? Also, something doesn’t add up here. Either the estimates in the equation or PTR values are wrong. For example: When the mean value of male, incisor and PTRS (40.64) is used in the first equation (Age = 67.530 - 207.669 (PTRS)), the estimated age is coming up to be less than zero (-8372.13). So, there is a definitely a miscalculation or typo error somewhere.
· Line 205-206: The numbers seem to be contradicting the text. It is also little confusing what is being considered high vs low because of the negative sign. So, would recommend rephrasing those sentences to make it clear may be use the terms: stronger negative correlation instead.
· Line 235: “The results of the study found that the incisors' overall mean PTR was higher than the canines.” More analysis may be needed to confirm if the difference is statistically significant. May need t-test to confirm the significant difference.
· Line 250-251: “In this study, measurements of PTR showed no significant gender difference” No analysis or results were shared comparing the difference between PTR between genders
· Line 274: Missing full form of PrRR
· Line 315: The higher(closer) R2 value to 1 is preferred in model selection. But in the current sentence it sounds like low R2 value is preferred over high R2 value. So, may have to be rephrased
· Line 317: Missing full form of SER
Conclusions:
· Line 351-352: “The correlation coefficients were better for maxillary incisors in both gender and planes compared to canines” But there were statistical tests performed/presented to see if these differences were statistically significant.
· Line 352-353: “We found that including gender as a predictor did not improve the precision of age estimation.” No mention of this anywhere in the methods and results. How was gender included in the model? Was it tested in an univariate model or multivariate model?
· Line 357: “acceptable range set by forensics i.e., SE of ten years.” Please provide the citation of the “forensics” that recommends the SE of ten years in the methods section. Not clear what is being referred here by mentioning as forensics? Any previous studies? If so, please cite them.
References:
The following study could be used in introduction or discussion
· Alharbi HS Sr, Alharbi AM, Alenazi AO, Kolarkodi SH, Elmoazen R. Age Estimation by Kvaal's Method Using Digital Panoramic Radiographs in the Saudi Population. Cureus. 2022 Apr 2;14(4):e23768. doi: 10.7759/cureus.23768. PMID: 35509748; PMCID: PMC9060989.
· Nagarajappa AK, Alruwaili MG, Alrubiash AAA, Alam MK. Adult Age Estimation from Dental Pulp in Jouf Population: A Digital Radiographic Study. World J Dent 2018;9(6):476-480.
Minor rephrasing may be needed for better sentence structuring.
Reviewer 4 Report
The manuscript is entitiled "Relationship Between Pulp Tooth Area Ratio and Chronological Age among Saudi Arabian Adults: A CBCT image analysis"
Major concerns:
Very limited sample size.
The manuscript focuses primarily on the use of pulp tooth ratio and CBCT imaging for age estimation, but it does not discuss or compare these methods with other established techniques in the field of forensic odontology.
While the manuscript briefly mentions some limitations, such as the lack of restriction to one quadrant of the mouth and the absence of information on systemic diseases and orthodontic treatment, it does not thoroughly discuss other potential limitations inherent to the study design or methodology.
The manuscript does not discuss the potential practical implications or applications of the findings in forensic settings or other relevant fields.
Minor concerns:
The introduction provides a general overview of the importance of age estimation. However, it would be helpful to provide a more specific context for the study. Why is age estimation particularly important in the Saudi Arabian population? Are there specific applications or challenges that make this research relevant?
Several studies are cited to support the use of PTR for age prediction, including studies conducted in Egyptian, Japanese, and Belgian populations. While these studies provide evidence for the value of PTR, it would be helpful to summarize their main findings and highlight any limitations or gaps in the existing research.
It was mentioned that the lack of studies conducted in the Saudi Arabian population using CBCT to correlate PTR of anterior teeth with chronological age. It would be helpful to explain why studying the Saudi Arabian population is important and how the findings of this study can contribute to the existing knowledge in the field.
The description of the measurement procedure is clear, and the provided figures help in understanding the process. However, it would be useful to mention how the examiners ensured consistency and reproducibility in their measurements. Were any intra- or inter-examiner reliability tests conducted?
Round 2
Reviewer 3 Report
The study looks much better after the corrections. However, few minor corrections or clarifications are needed.
Specific comments:
· Section 2.4 Statistical analysis: Some of the sentences were repeated twice. Please verify
· Please add p-values of t-tests comparing the variables across teeth (table 1) as well to support the statement in line 285-286 “The results of the study found that the incisors' overall mean PTR was higher than the canines in axial plane only.”
· Line 411-412: “We found that including gender as a predictor did not improve the precision of age estimation” – Based on the author response to previous conmment it looks like it needs to be rephrased but it was missed. So, please verify and rephrase accordingly
· For figure 1 & 2, would recommend adding the PTR ratio in the description or as a legend
· Some of the decimals in tables were rounded to two digits and few were more than 2 digits. Would recommend to keep them consistent as per author guidelines.
Reviewer 4 Report
The authors addressed all the comments; therefore, the manuscript can be published.
